# Propagated infra-slow intrinsic brain activity reorganizes across wake and slow wave sleep

Anish Mitra[1]*, Abraham Z Snyder[1,2], Enzo Tagliazucchi[3,4], Helmut Laufs[4,5], Marcus E Raichle[1,2]

[1]Department of Radiology, Washington University in St. Louis, St. Louis, United States; [2]Department of Neurology, Washington University in St. Louis, St. Louis, United States; [3]Institute for Medical Psychology, Christian-Albrechts-Universität zu Kiel, Kiel, Germany; [4]Department of Neurology, Brain Imaging Center, Goethe-Universität Frankfurt am Main, Frankfurt, Germany; [5]Department of Neurology, Christian-Albrechts-Universität zu Kiel, Kiel, Germany

**Abstract** Propagation of slow intrinsic brain activity has been widely observed in electrophysiogical studies of slow wave sleep (SWS). However, in human resting state fMRI (rs-fMRI), intrinsic activity has been understood predominantly in terms of zero-lag temporal synchrony (functional connectivity) within systems known as resting state networks (RSNs). Prior rs-fMRI studies have found that RSNs are generally preserved across wake and sleep. Here, we use a recently developed analysis technique to study propagation of infra-slow intrinsic blood oxygen level dependent (BOLD) signals in normal adults during wake and SWS. This analysis reveals marked changes in propagation patterns in SWS vs. wake. Broadly, ordered propagation is preserved within traditionally defined RSNs but lost between RSNs. Additionally, propagation between cerebral cortex and subcortical structures reverses directions, and intra-cortical propagation becomes reorganized, especially in visual and sensorimotor cortices. These findings show that propagated rs-fMRI activity informs theoretical accounts of the neural functions of sleep.

*For correspondence:
mikumusic@gmail.com

**Competing interests:** The authors declare that no competing interests exist.

## Introduction

Sleep is a state during which interactions with the environment are greatly attenuated. The behavioral consequences of this state, namely, immobility and reduced responsiveness, carry obvious costs, such as compromised avoidance of predators. Nevertheless, nearly all animals sleep, suggesting that sleep is essential to normal physiology (*Cirelli and Tononi, 2008*). In most mammals, including humans, prolonged sleep deprivation leads to impaired performance, psychosis, and eventually death (*Brown et al., 2012*; *Everson et al., 1989*; *Rechtschaffen, 1998*). Sleep is attended by changes in gene expression (*Abel et al., 2013*; *Cirelli and Tononi, 2000*), neuromodulator levels (*Brown et al., 2012*), metabolism (*Boyle et al., 1994*; *Braun et al., 1997*), and markedly altered patterns of neural activity (*Dang-Vu, 2012*; *Loomis et al., 1935*; *McCormick and Bal, 1997*). Yet, the fundamental functions of sleep remain elusive.

Sleep conventionally is divided into stages, the deepest of which is slow wave sleep (SWS; also known as N3 sleep) (*Soeffing et al., 2008*; *Rechtschaffen and Kales, 1968*). The electrophysiologic hallmark of SWS is the slow oscillation, which manifests as periodic alternations of membrane potential, also known as Up/Down states (UDSs), characteristically at frequencies in the range of 0.5–1.5 Hz (*Achermann and Borbély, 1997*; *Steriade et al., 1993*). As observed with extracellular local field potential (LFP) recordings, slow oscillations are locally synchronous but exhibit apparent propagation

**eLife digest** The brain shows spontaneous activity all the time, even when we are sleeping. A technique called functional magnetic resonance imaging (fMRI) has revealed that this spontaneous activity can occur in distinct groups of brain regions at roughly at the same time. Each group is referred to as a resting-state network and the brain regions that make up these networks are largely the same between individuals, and between the sleep and awake states.

However, when spontaneous brain activity is measured in rodents and humans using electrodes, it appears that there are actually waves of electrical activity that spread both within and across resting-state networks. In other words, these studies suggest that brain regions tend to become active in turn rather than at the same time. This led Mitra et al. to question whether the techniques used to analyze fMRI scans of spontaneous brain activity might have overlooked differences in the timing of brain activity.

Mitra et al. used a new technique to analyze fMRI data from healthy adult volunteers. The experiments show that brain regions are activated in a different order depending on whether the individuals are awake or asleep. Specifically, in conscious individuals information from the senses is first processed by a structure deep within the brain called the thalamus before it is passed to the brain's outer layer, known as the cortex. During deep sleep, this flow of information is reversed and signals are instead sent from the cortex to the thalamus. This may contribute to our loss of sensory awareness during sleep, and even to the occurrence of dreaming.

The exchange of informationbetween resting-state networks also becomes disorganized during sleep. This lends support to the idea that the coordinated transfer of information between networks in the awake state may contribute to consciousness. Future experiments should explore differences in spontaneous brain activity in different phases of sleep, and investigate how such activity is able to spread throughout the brain.

over the whole brain on a time scale of 100's of milliseconds (*Hahn et al., 2012*; *Nir et al., 2011*; *Riedner et al., 2007*; *Sheroziya and Timofeev, 2014*). Similar patterns of propagation have also been observed using electroencephalography (EEG), in which SWS manifests as a predominance of high amplitude, 1–4 Hz (delta) rhythms (*Massimini et al., 2004*).

The observation of propagated slow electrophysiological activity during SWS raises the question of whether similar phenomenology might be observed by other means, specifically, resting state functional magnetic resonance imaging (rs-fMRI). Infra-slow (<0.1 Hz) intrinsic (equivalently, spontaneous) activity recorded using rs-fMRI has been understood predominantly in terms of zero-lag temporal synchrony (functional connectivity) within systems known as resting state networks (RSNs) (*Beckmann et al., 2005*; *Biswal et al., 2010*). Prior rs-fMRI studies have found that RSNs are generally preserved across wake and SWS (*Horovitz et al., 2009*; *Larson-Prior et al., 2009*; *Picchioni et al., 2013*; *Sämann et al., 2011*; *Tagliazucchi et al., 2013a*). Importantly, conventional functional connectivity analyses assume temporal synchronicity and make no provision for the possibility that intrinsic activity may propagate between regions.

We have recently described an alternative analysis technique which explicitly focuses on apparent propagation in rs-fMRI data (*Mitra et al., 2014*; *2015*; *Yuste and Fairhall, 2015*). Our methodology applies parabolic interpolation to lagged cross-covariance curves to detect temporal lags at a resolution finer than the temporal sampling density of rs-fMRI (see Materials and methods; *Figure 1*). Using this technique, we previously demonstrated, in awake, normal humans, that the blood oxygen level dependent (BOLD) signal exhibits highly reproducible temporal lag patterns on a time scale of ~1 s; some regions are systematically early with respect to the rest of brain, whereas other regions are systematically late (*Mitra et al., 2014*; *2015*). Moreover, temporal lags in BOLD signal activity are altered, with appropriate focality, by prior performance of motor tasks as well as by time of day (*Mitra et al., 2014*). We operationally infer apparent propagation on the basis of measured temporal lags, assuming nothing regarding the path or mechanism by which BOLD signals 'propagate' between regions. In particular, the temporal scale of this phenomenology is much slower than axonal transmission via fiber tracts (*Caminiti et al., 2009*). With this understanding, we omit 'apparent' in references to BOLD signal propagation.

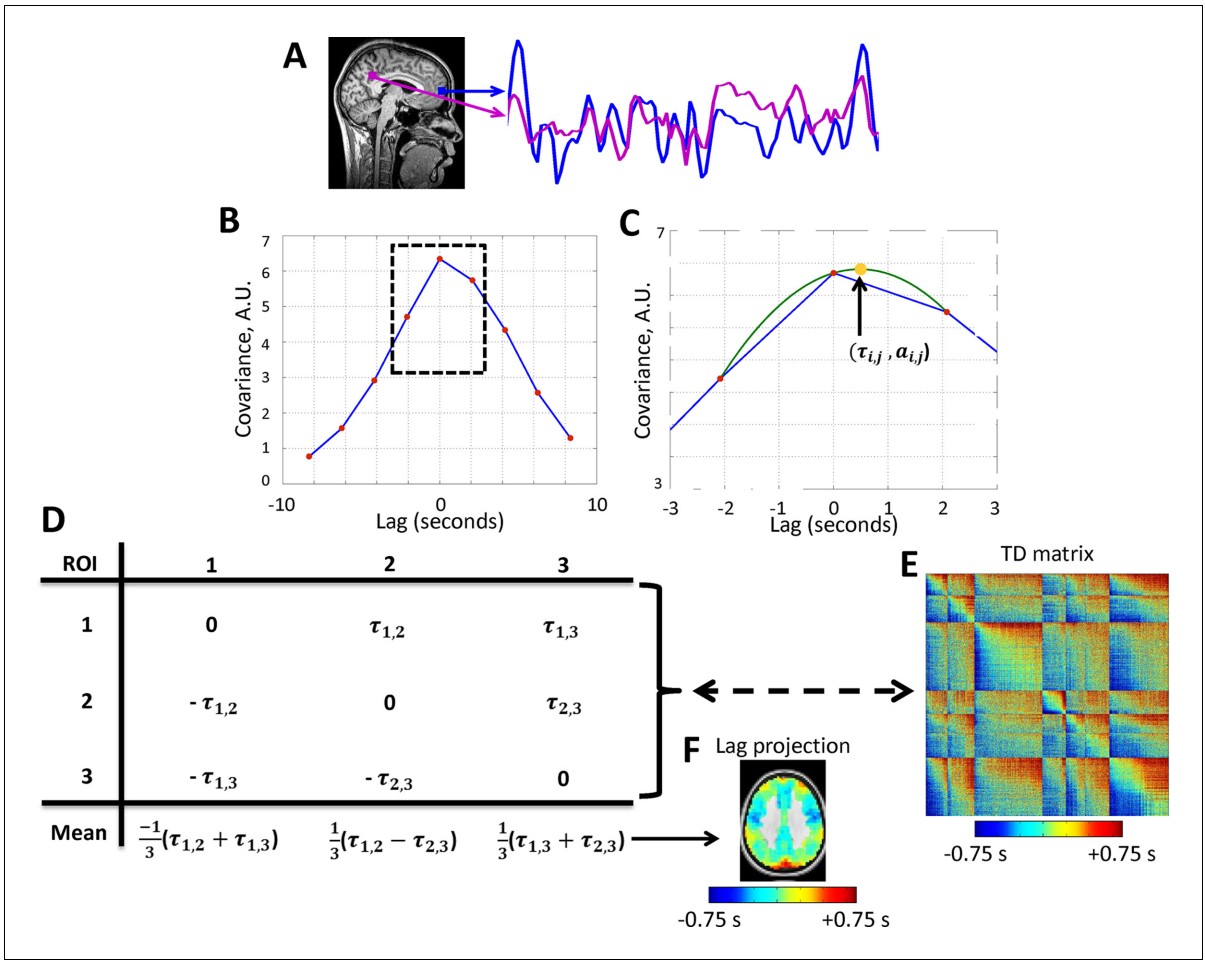

**Figure 1.** Calculation of lag structure using lagged cross-covariance functions and parabolic interpolation. Lags are defined by analysis of timeseries derived from two loci. (**A**) Two exemplar loci (both in the default mode network). The time series were extracted from the illustrated loci over ~200 s. (**B**) The corresponding lagged cross-covariance function (Materials and methods *Equation 2*). Although the lagged cross-covariance is defined over the range ±T, where T is the run duration, the range of the plotted values is restricted to ± 8.32 s, which is equivalent to ± 4 frames (red markers) as the repetition time was 2.08 s. The lag between the time series is the value at which the absolute value of the cross-covariance function is maximal. (**C**) This extremum can be determined at a resolution finer than the temporal sampling density by parabolic interpolation (green line) through the computed values (red markers). This extremum (arrow, yellow marker) defines both the lag between time series i and j ($\tau_{i,j}$) and the corresponding amplitude ($a_{i,j}$). (**D**) Toy case illustration of a time-delay (TD) matrix (Materials and methods *Equation 3*) representing 3 voxels. TD matrices encode the lag between every pair of analyzed voxels and are anti-symmetric by definition. The mean over each column of a TD matrix generates a lag projection map (Methods *Equation 4*). A TD matrix obtained with real rs-fMRI data and the corresponding lag projection map are shown in panels (**E**) and (**F**), respectively. Panels A-D are adapted from *Mitra et al., 2014*.

Previous work has related propagated slow electrophysiological activity during SWS to several aspects of physiology. First, SWS is believed to represent an off-line state during which propagated slow activity plays a central role in the consolidation of memory (*Born et al., 2006*; *Maquet, 2001*; *Marshall et al., 2006*; *Stickgold, 2005*). In particular, propagation along the anterior-posterior axis of the brain has been linked to consolidation of declarative memory (*Marshall et al., 2006*). Second, loss of environmental awareness during SWS is thought to be mediated by thalamic hyperpolarization, which is modulated by propagation of slow (<1 Hz) activity from cortex to thalamus (*Blethyn et al., 2006*; *Hughes et al., 2002*). Evidence of such propagation has never been observed in vivo. Finally, it has been theorized that reduced subjective awareness ('unconsciousness') during SWS results from loss of integration across networks (*Mashour, 2005*; *Tononi, 2004*). Here, we report whole-brain patterns of propagated rs-fMRI activity in humans, contrasting eyes-closed wake vs. SWS. Our results lend support to each of the aforementioned perspectives.

# Results

We characterize lag structure (e.g., apparent propagation) using three major approaches. First, we begin by computing lags for all pairs of voxels in gray matter in rs-fMRI data (*Figure 1*; *Mitra et al., 2014*). These results are assembled into *time-delay (TD) matrices*, which have dimensions voxels × voxels and entries in units of seconds (Materials and methods *Equation 3*). TD matrices represent the lag between all pairs of voxels in gray matter. Second, computing the mean over all columns of the TD matrix yields a *lag projection map* (Materials and methods *Equation 4*; *Figure 1*). Lag projection maps topographically represent the mean lag between each voxel and the rest of the brain. Third, computing lags over the whole brain with respect to a particular region yields a *seed-based lag map*. Seed-based lag maps topographically represent the degree to which each voxel is, on average, early vs. late with respect to the selected seed. The present results are reported in terms of lag projection maps (*Figure 2*), seed-based lag maps (*Figure 3* and *4*), and TD matrices (*Figure 5*). Additionally, it is possible to decompose lag structure into multiple temporal sequences ('*lag threads*') by applying spatial principal components analysis (PCA) to the TD matrix (*Mitra et al., 2015*) (see *Figure 7* caption for details). Lag thread results are presented in *Figure 7*.

*Figure 2A, B* exhibit lag projection maps computed during wake and SWS. State-dependent shifts are evident in the lag projection maps, for example in occipital cortex and thalamus (*Figure 2A, B*). *Figure 2C* shows all statistically significant spatial clusters ($|Z| > 4.5$, $p < 0.05$ corrected; see Materials and methods) in the wake vs. SWS comparison. These clusters include: thalamus, bilateral putamen, brainstem, visual cortex, medial prefrontal cortex (mPFC), and paracentral lobule (PCL) (*Figure 2C*). Visual cortex was later (more positive lag values) during wake as compared to SWS, whereas the remaining clusters were earlier (more negative lag values).

Having found voxel clusters exhibiting statistically significant changes in lag structure in the whole-brain wake vs. SWS contrast, we next computed seed-based lag maps using the clusters shown in *Figure 2C* as seeds (see Materials and methods). Seed-based lag maps represent temporal lags between each voxel and the average timecourse computed over the seed-region of interest.

Seed-based lag maps obtained with the subcortical clusters, specifically, thalamus, putamen, and brainstem, are shown in *Figure 3*, which illustrates altered lags during SWS compared to wake. Three principal findings are evident. First, whereas cortex is generally late with respect to the subcortical seeds during wake, cortex becomes earlier than subcortical structures during SWS (see significant differences in *Figure 3*). Second, a 'front-to-back' propagation pattern appears in SWS; this phenomenon is best seen in the sagittal views of the thalamus and putamen lag maps (pink arrows, *Figure 3A, B*). This pattern may correspond to previous reports of slow wave propagation along the anterior-posterior axis of the brain (*Massimini et al., 2004*; *Murphy et al., 2009*). Third, the lag structure within the thalamus and brainstem (*Figure 3A,C*, pink ovals) remains largely constant across states. An early-to-late sequence extending from lower brainstem to rostral thalamus is evident in each of the seed-based lag maps shown in *Figure 3A,C* (pink ovals). Hence, the general pattern of BOLD signal propagation between cortex and subcortical structures reverses during SWS, but propagation within the brainstem-thalamus axis is largely preserved across wake vs. SWS. A broader view of these results is shown in *Figure 3—figure supplement 1* and *2*.

*Figure 4* displays seed-based lag maps obtained with the cortical clusters shown in *Figure 2C*. State-contrasts in lag structure differ by seed. Specifically, visual cortex (*Figure 4A*) is neither wholly late nor early during wake, but nearly the entire cerebral cortex becomes late with respect to visual cortex during SWS. Two especially prominent foci of lateness in SWS are dorsolateral prefrontal cortex and the paracentral lobule (*Figure 4A*). A variety of lag shifts are evident in the results obtained with the medial prefrontal seed (*Figure 4B*). For example, subgenual prefrontal cortex (red arrow) shifts from mid-latency (lag values near zero) during wake to very early during SWS, and the paracentral lobule shifts from mid-latency during wake to late during SWS. A 'front-to-back' propagation pattern (also highlighted in *Figure 3*) is clearly evident in the sagittal view in *Figure 4B* during SWS. *Figure 4C* illustrates dramatic changes in the lag relations of the paracentral lobule. In wake, paracentral lobule leads both lateral sensory-motor cortex and posterior insula. These relations are reversed in SWS, during which nearly the entire cortex becomes markedly early with respect to the paracentral lobule. A broader view of these results is shown in *Figure 4—figure supplement 1* and *2*.

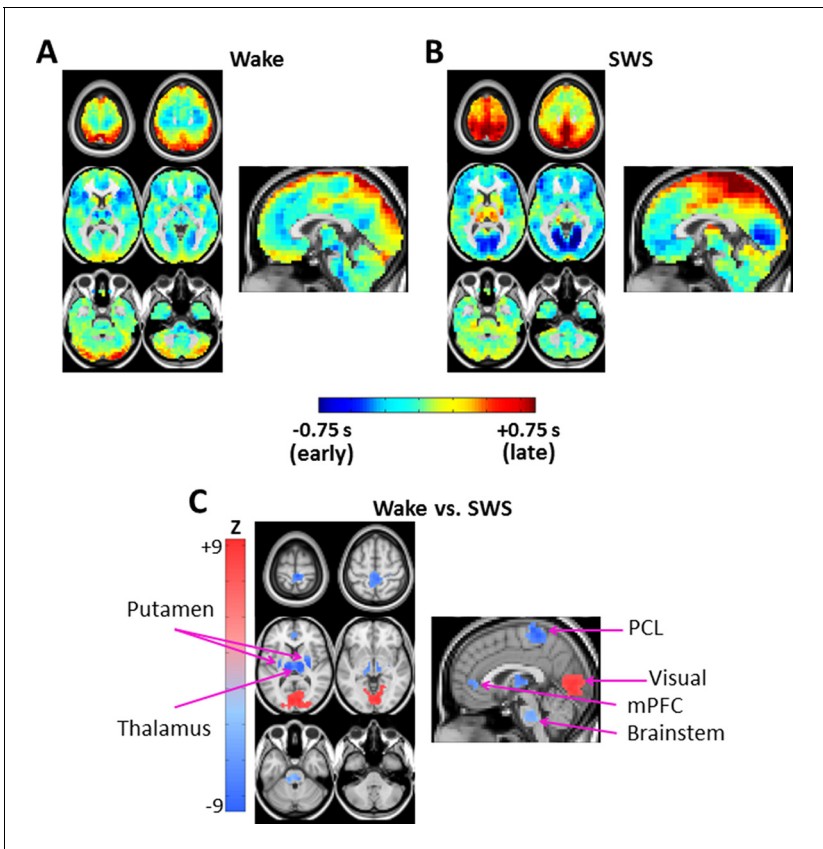

**Figure 2.** Lag projection maps in wake and slow wave sleep. Lag projection maps depict the mean lag between each voxel and the rest of the brain (*Mitra et al., 2014*; *Nikolić, 2007*). Panels A-B display lag projection maps, in units of seconds, derived from wake (**A**) and SWS (**B**). These lag projection maps demonstrate changes in lag structure as a function of state. For example, thalamus is early (blue) during wake but late (red) during SWS; the opposite shift is evident in visual cortex. Panel C shows voxels exhibiting cluster-wise statistical significance in the wake vs. SWS comparison ($|z| > 4.5$, p<0.05 corrected). These clusters are centered on putamen bilaterally, thalamus, paracentral lobule (PCL), visual cortex, medial prefrontal cortex (mPFC), and brainstem. Axial slices: Z = +69, +57, +9, -3, -27, -39. Sagittal slice: X = +3.

The results presented so far highlight topographic features of apparent propagation that change between wake and SWS. The next set of results considers pair-wise lag relations defined over pairs of 6mm$^3$ isotropic cortical gray matter voxels. The results are presented as time-delay (TD) matrices. Voxels in the TD matrices are ordered first by resting state network affiliation (see *Figure 5—figure supplement 2*). Then, within RSNs, the voxels are ordered from earliest (most negative) to latest (most positive) according to mean latency determined in wake (*Figure 5A*). The ordering computed during wake was applied to the SWS TD results (*Figure 5B*). A key algebraic feature of TD matrices is that they are exactly anti-symmetric ($\tau_{i,j} = -\tau_{j,i}$; Materials and methods *Equation 3*). Thus, if the lag between voxels $i$ and $j$ is $\tau$ seconds, then the lag between voxels $j$ and $i$ must be exactly $-\tau$ seconds (see Materials and methods).

As TD matrices are anti-symmetric, each diagonal block, which represents intra-RSN lag structure, is anti-symmetric as well. However, the algebra does not impose any relation between lag and RSN membership. Thus, the structure evident in the wake TD matrix (*Figure 5A*) is informative, and recapitulates previous findings obtained in a separate, large data set (*Mitra et al., 2014*). The diagonal blocks show a wide range and well-ordered distribution of lags. This organization reflects propagation within RSNs; the DMN block, highlighted in green in *Figure 5A*, is an example of intra-network lag organization. The off-diagonal blocks, which represent lags across RSNs, also contain well-ordered early, middle, and late components, much like the diagonal blocks. This feature is not algebraically imposed. An important implication of the early-to-late organization of the inter-RSN blocks in *Figure 5A* is that each RSN is neither early nor late with respect to the others during the wake

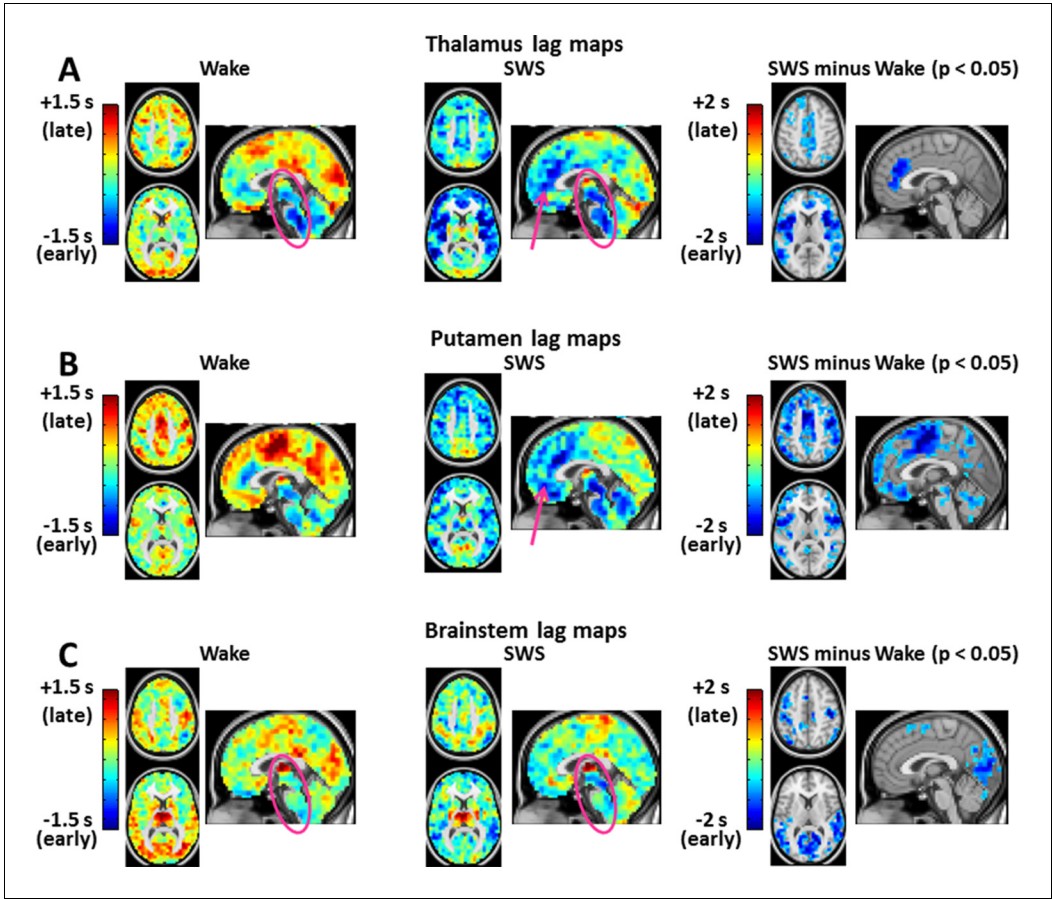

**Figure 3.** Seed-based lag maps in wake and SWS corresponding to the subcortical regions identified in *Figure 2C*: thalamus (**A**), putamen (**B**), and brainstem (**C**). Also shown are lag difference maps (SWS minus wake) thresholded for cluster-wise statistical significance ( Z > 4.5, p<0.05 corrected; as in *Figure 2C*). During wake, the cerebral cortex is generally late (yellow/red hues) with respect to subcortical regions. The cerebral cortex becomes early (blue/green hues) with respect to subcortical areas during SWS. All significant lag differences are negative (blue), and predominantly found in cortex. Pre-frontal cortex becomes markedly early with respect to the posterior parts of the brain (pink arrows in A and B). This feature suggests a 'front-to-back' propagation of slow waves in SWS (*Massimini et al., 2004*). Lag structure in the brainstem and thalamus is relatively constant (pink ovals in A and C). Lag structure is present within the thalamus in panel A even though the whole thalamus was used as the reference seed. This effect is observed because voxels within large seed-regions, e.g., thalamus, can exhibit non-zero lags with the mean timecourse computed over the entire seed. Axial slices: Z = +45, +9. Sagittal slice: X = +3. An expanded view of these results is shown in *Figure 3—figure supplement 1* and *2*.

The following figure supplements are available for figure 3:

**Figure supplement 1.** Expanded view of seed-based lag maps derived using thalamus, putamen, and brainstem regions of interest defined in *Figure 2C*.

**Figure supplement 2.** Seed-based lag difference maps thresholded for cluster-wise statistical significance of p<0.05 (as in *Figure 2*).

state, a feature which we have previously reported (*Mitra et al., 2014*). To illustrate this point, consider the off-diagonal block corresponding to the DMN paired with the dorsal attention network (DAN), outlined in yellow in *Figure 5A*. A well-ordered progression from early (blue) to late (red) is evident, indicating that parts of the DMN lead the DAN and *vice versa*; neither the DMN nor the DAN leads or follows the other as a whole.

We next compared the TD matrix in SWS (*Figure 5B*) to the TD matrix in wake. Three features emerged from this comparison. First, the range of lag values in SWS is larger than in the wake data, as if the speed of BOLD signal propagation has slowed. We quantitatively represent this effect as the standard deviation (SD), computed over the upper triangular (hence unique) lag values in wake

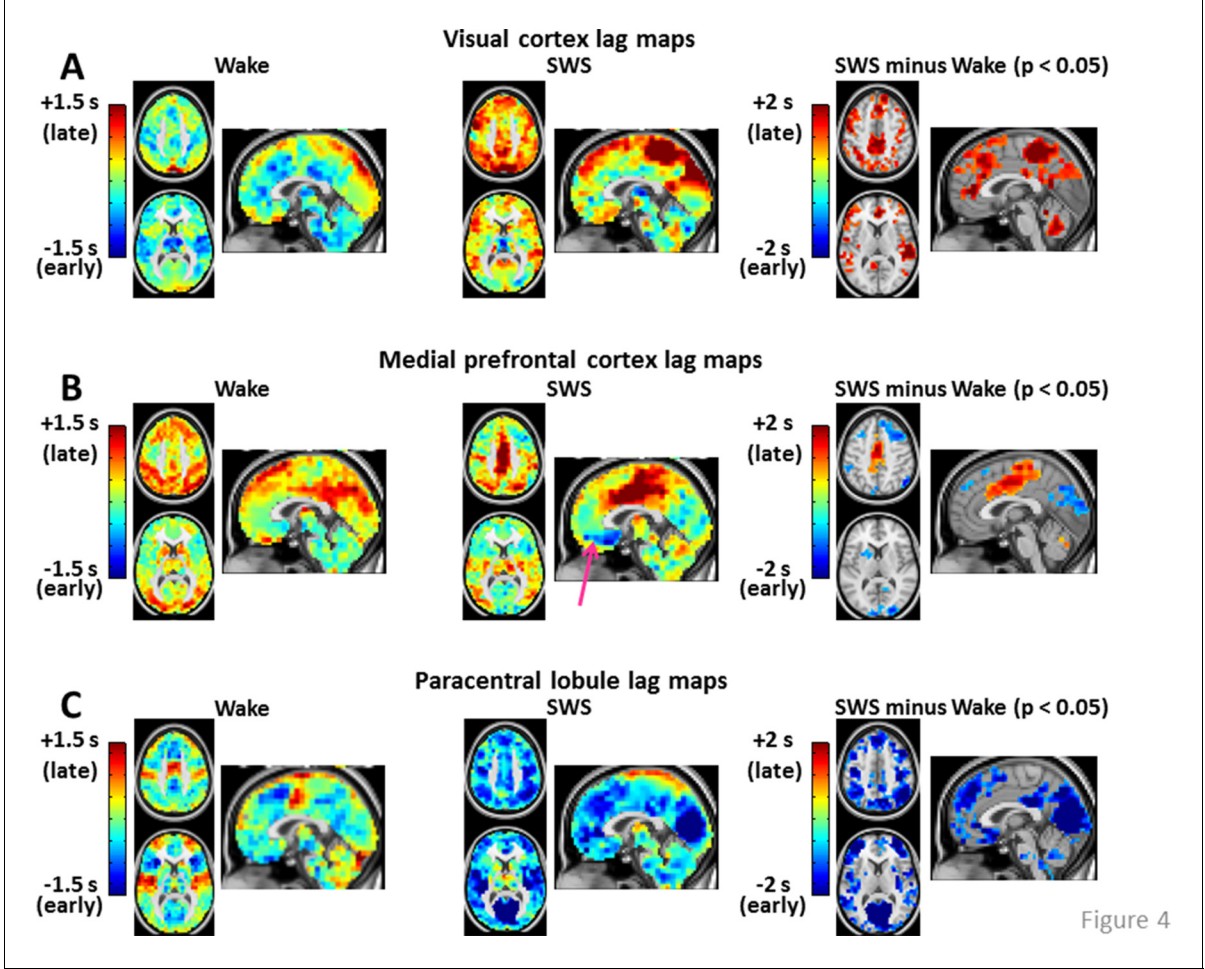

**Figure 4.** Seed-based lag maps in wake and SWS corresponding to the cortical regions identified in **Figure 2C**: visual cortex (**A**), medial prefrontal cortex (**B**), and paracentral lobule (**C**). Also shown are lag difference maps (SWS minus wake), thresholded for statistical significance, as in **Figure 2**. Panel A shows that, whereas the visual seed is neither wholly late nor early in wake, nearly the entire cortex is late with respect to visual cortex during SWS. Panel B shows that medial prefrontal cortex (mPFC) exhibits both early and late lag shifts between wake and SWS. The mPFC lag map in SWS also exhibits the 'front-to-back' propagation pattern highlighted in **Figure 3** (pink arrow). Panel C shows that many of the lag relations of the paracentral lobule seed are reversed during SWS relative to wake. For example, in wake, the seed-region leads lateral sensory-motor cortex and posterior insula. These relations reverse in SWS and nearly the entire cerebral cortex becomes early with respect to the paracentral lobule. An expanded view of these results is shown in **Figure 4—figure supplement 1** and **2**. Slice coordinates identical to **Figure 3**.

The following figure supplements are available for figure 4:

**Figure supplement 1.** Expanded view of seed-based lag maps derived using visual, medial prefrontal cortex, and paracentral lobule regions of interest defined in **Figure 2C**.

**Figure supplement 2.** Seed-based lag difference maps thresholded for cluster-wise statistical significance (as in **Figure 2**).

and SWS. SD during wake was 0.42 ± 0.05 s, whereas SD during SWS was 0.65 ± 0.07 s. This effect was significant (p<0.01, by permutation resampling with 10,000 trials).

Second, inter-RSN lag structure is altered in SWS. This effect appears as a loss of early (blue) to late (red) organization in off-diagonal blocks, e.g., as in the DAN:DMN block (yellow outline in **Figure 5B**). To quantitatively assess this change, we computed the rank correlation (Spearman's $\rho$) between wake and SWS lag values over all voxel pairs in each of the $8 \cdot 7/2 = 28$ unique intra- and inter-RSN blocks (**Figure 5A, B**). Blocks exhibiting a significantly low correlation (p<0.05, corrected for 28 multiple comparisons) comparing wake vs. SWS are marked with a white asterisk in **Figure 5C**; these effects represent a significant change in lag structure between resting state

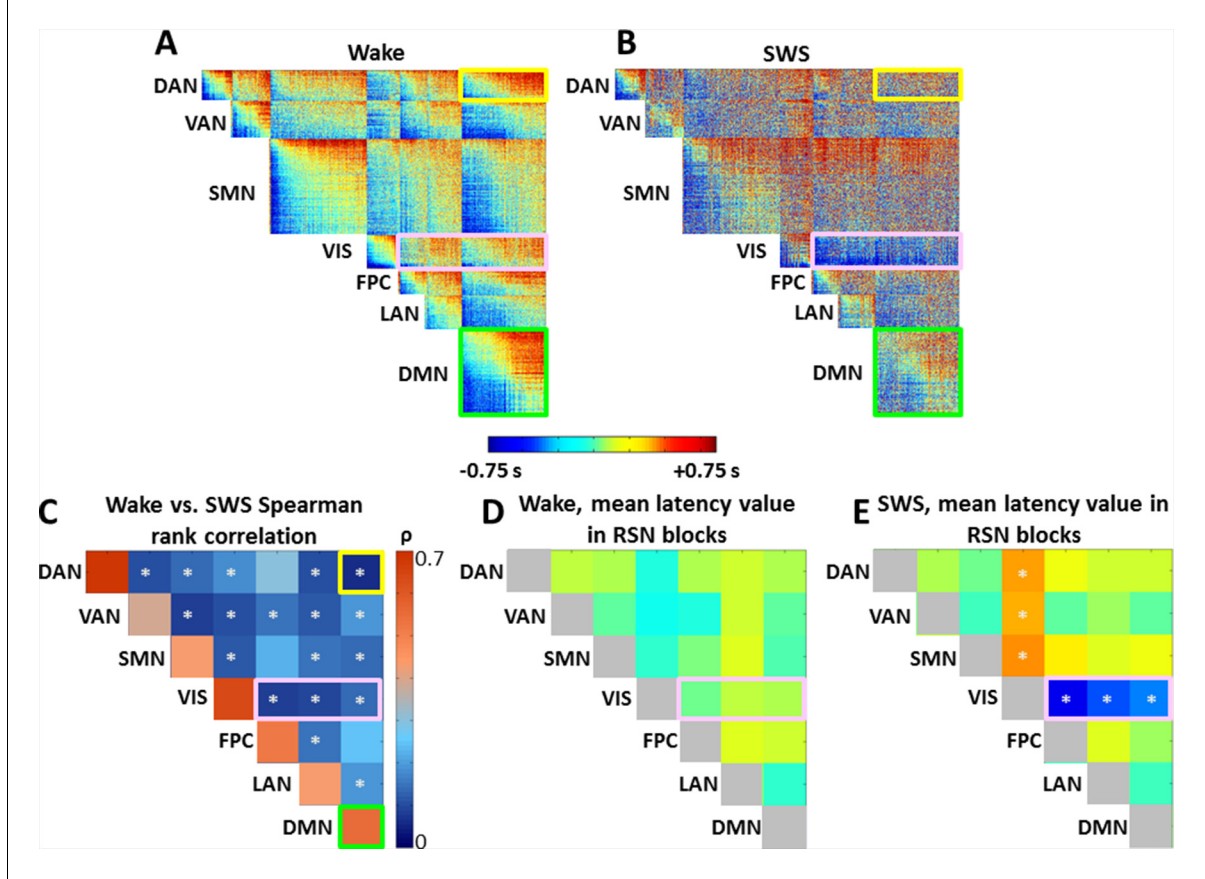

**Figure 5.** Time delay (TD) matrices. Panels A-B display TD matrices (in units of seconds) in wake and SWS, respectively. Each pixel represents the lag between two voxels. TD matrices are, by definition, anti-symmetric. Hence, all relevant information is contained in the displayed upper triangular values. The matrices displayed in (A-B) have been masked to include only cortical (6 mm)$^3$ voxels with a ≥90% chance of belonging to one of 7 resting state networks (RSNs: dorsal attention network (DAN), ventral attention network (VAN), sensory motor network (SMN), visual network (VIS), frontoparietal control network (FPC), language network (LAN), default mode network (DMN)) ((*Hacker et al., 2013*), see *Figure 5—figure supplement 2*). The same RSN definitions are applied in wake and SWS. After sorting voxels by RSN membership, voxels were sorted from early to late within RSN in the wake state. The wake ordering was applied to the SWS TD matrix. Panel C shows the Spearman rank order correlation between wake and SWS lag values in each of the 28 intra- and inter- RSN blocks in panels (A-B). The diagonal blocks in panel C exhibit high correlation values, indicating that intra-RSN propagation is relatively preserved across wake and SWS. The correlation values in the off-diagonal blocks in panel C are low, demonstrating that inter-RSN propagation is strongly altered during SWS. White asterisks indicate significant (p<0.05) effects computed by permutation resampling, including correction for multiple (N = 28) comparisons. Panel D plots the mean within-block values of the wake TD matrix shown in panel A. The values in panel D (in units of sec) are very near zero, implying that no RSN is entirely leads or follows other RSNs. Panel E plots the mean within-block values of the SWS TD matrix shown in panel B. Note significant visual network earliness with respect to other networks (p<0.05 by permutation resampling, multiple comparisons corrected for 21 upper diagonal blocks). Owing to anti-symmetry, the horizontal blue and vertical orange blocks both represent visual earliness. Diagonal blocks in panels D and E are colored gray to symbolize that they are constrained to be zero-mean. TD matrices with alternate voxel orderings are shown in *Figure 5—figure supplement 1*.

The following figure supplements are available for figure 5:

**Figure supplement 1.** Time delay (TD) matrices as a function of sleep stage, alternate voxel orderings.

**Figure supplement 2.** Resting state network (RSN) assignments for gray matter voxels used to compute the TD matrix and functional connectivity matrix results shown in main *Figures 5*, *6.*

networks. Importantly, these effects were observed only in off-diagonal (cross-RSN) blocks. In other words, intra-network lag structure was relatively preserved ($\rho > 0.6$) across states, e.g., within the DMN (green outline *Figure 5A, B*). In contrast, inter-network lag structure was markedly altered ($\rho < 0.3$) in the majority of off-diagonal blocks. Thus, *Figure 5C* demonstrates that, whereas

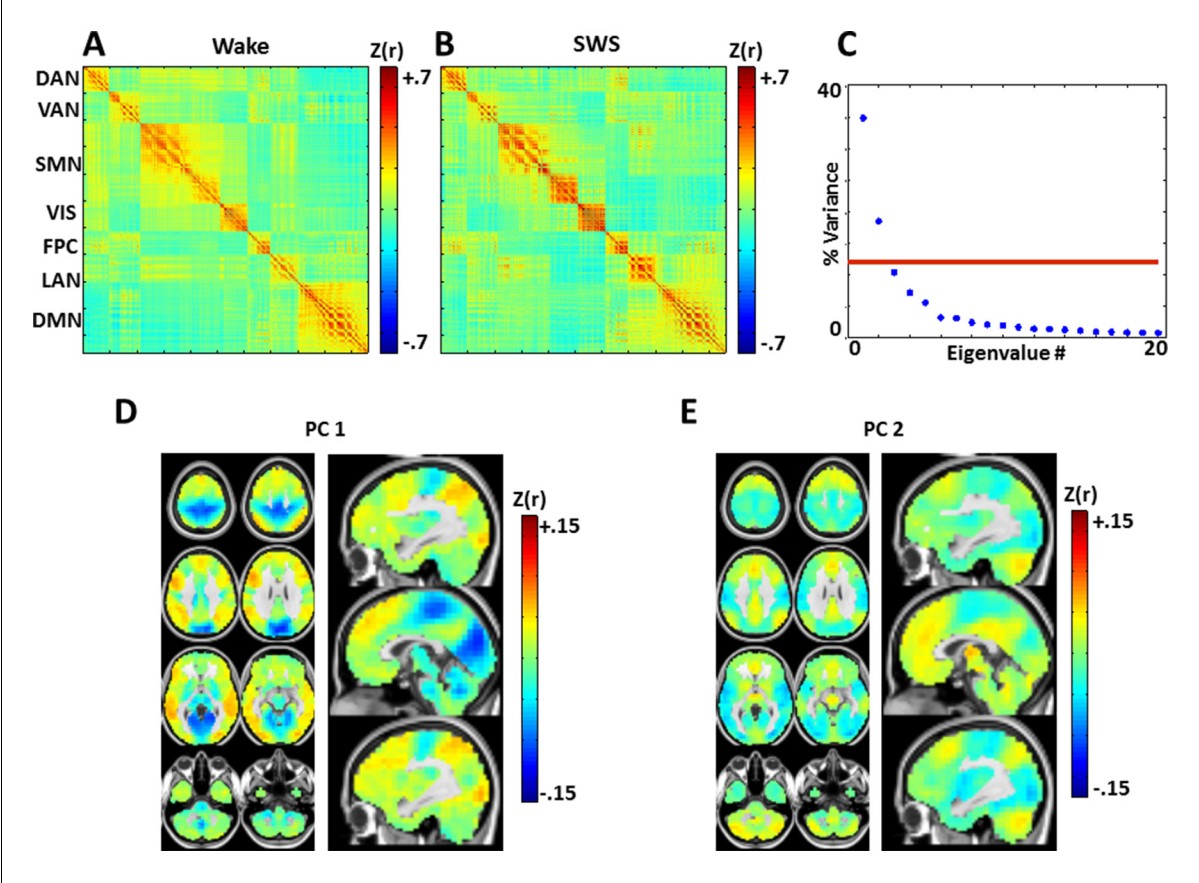

**Figure 6.** Zero-lag correlation (conventional functional connectivity; FC) matrices. (**A**): wake. (**B**): slow wave sleep. Voxels shown in the correlation matrices correspond to *Figure 5A, B* (see also *Figure 5—figure supplement 2*), and matrix values are Fisher-z transformed Pearson correlations averaged over subjects. Note relatively preserved RSN organization across states, in line with previous analyses of these data (*Tagliazucchi et al., 2013a*). To assess the topography of pair-wise correlation changes, we computed the difference between the SWS and wake correlation matrices (wake minus SWS), and applied spatial principal components analysis (PCA) to the difference matrix. The resulting eigenspectrum (panel C) shows that there are 2 statistically significant ($p < 0.05$; red line) PCs (threshold computed by permutation re-sampling). The topographies of these PC's are shown in panels D and E. These topographies reflect modest FC reductions in visual/somatosensory networks in wake vs. SWS, and modest FC increases in the DMN and thalamus in wake vs. SWS. These results are in accordance with previous findings comparing wake vs. NREM sleep (*Horovitz et al., 2008*; *Picchioni et al., 2013*). Importantly, zero-lag correlation structure (panels **A-B**) is much more preserved across wake and SWS than is lag structure (*Figure 5*). Axial slices: Z = +69, +57, +45, +33, +9, -3, -27, -39. Sagittal slices: X = +3, +12, -12.

propagation *within* RSNs is relatively preserved during sleep, propagation *across* RSNs is significantly altered. During SWS, most cross-RSN blocks (excluding visual network pairs) develop a 'disorganized' structure, in which well-ordered propagation appears to be lost. This effect is not attributable to voxel ordering, as the voxels are ordered identically in the wake and SWS TD matrices (*Figure 5A*). The same disorganized appearance persists even if the SWS TD matrix ROIs are ordered according to their own stage-specific lag structure (*Figure 5—figure supplement 1*).

A third effect evident in *Figure 5A, B* is the appearance of an 'all early' organization in the visual network ('VIS', pink box) in SWS. That is, in half the TD matrix, all cross-RSN voxel pairs involving the visual network exhibit negative (early; blue) lag values in SWS, indicating that voxels in the visual network are earlier than nearly all other cortical voxels. Thus, the principle, which applies in wake, that no RSN leads or follows any other (*Mitra et al., 2014*), does not hold during SWS.

To quantitatively investigate changes in lag structure between the visual network and the rest of the brain, we computed the mean lag value for each off-diagonal (inter-RSN) block in wake and SWS (*Figure 5D and E* respectively). Anti-symmetry forces a mean lag value of zero within diagonal blocks. Thus, 21 unique blocks are considered in this analysis. If no RSN leads or follows any other in aggregate, the average lag value in each inter-RSN block should be zero. *Figure 5D* illustrates that

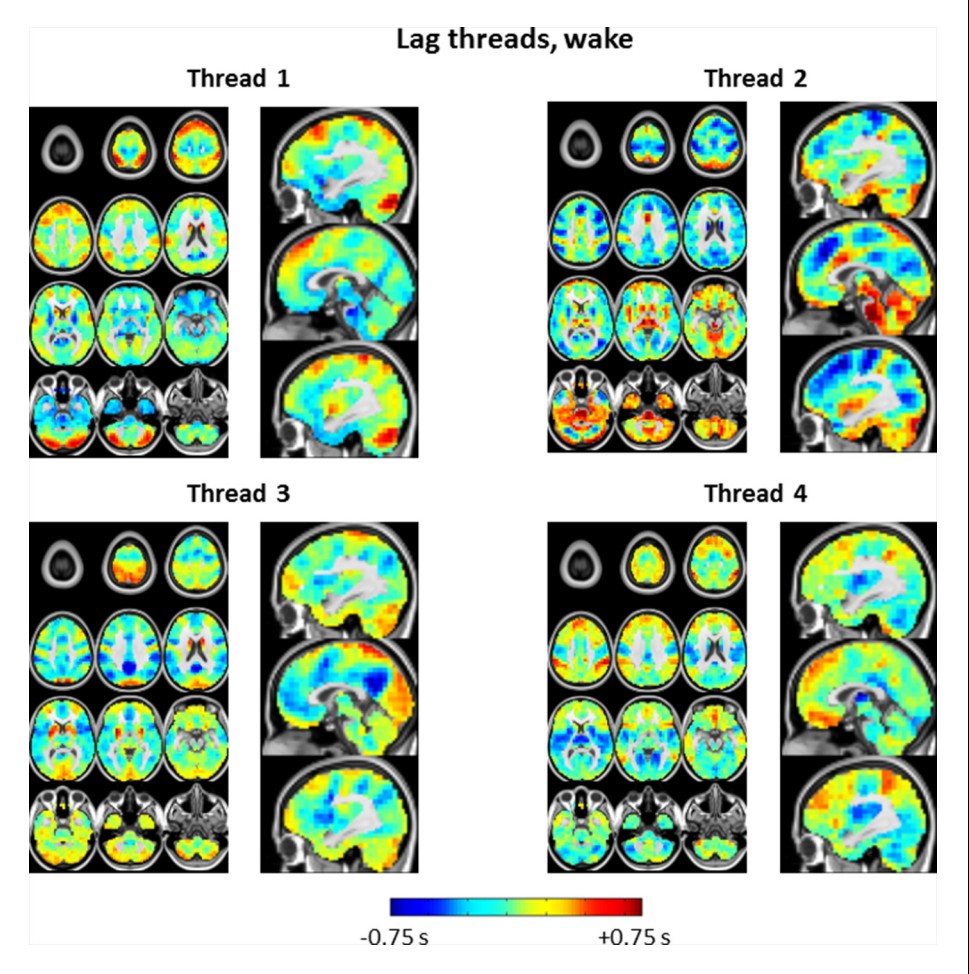

**Figure 7.** Lag structure dimensionality in wake and SWS. We have previously shown that multiple temporal sequences can be extracted from a TD matrix by applying spatial principal components analysis (PCA) to the TD matrix after zero-centering each column (*Mitra et al., 2015*). Here we show PCA-derived eigenvalues (scree plot) obtained in wake (pink) and SWS (blue). The maximum likelihood dimensionality estimates derived using the method of (*Minka, 2001*) are 4 and 3, respectively, in wake and SWS. The corresponding topographies ('lag threads') are shown in *Figure 7—figure supplement 1* and *2*. N.B.: We previously obtained a maximum likelihood TD dimensionality estimate of 8 in a much larger (N = 688) awake dataset (*Mitra et al., 2015*). The lower presently obtained figure (4) reflects less statistical power owing to a smaller subject sample (N = 39).

The following figure supplements are available for figure 7:

**Figure supplement 1.** Lag thread topographies in wake corresponding to the eigenvalues shown in *Figure 7*.

**Figure supplement 2.** Lag threads topographies in SWS.

this is the case during wake: the mean lag in each off-diagonal block is very nearly zero, as previously reported in a completely independent dataset (*Mitra et al., 2014*). *Figure 5E* shows that this principle approximately applies also in SWS *except in the visual network* (white asterisks). During SWS, voxels in the visual network become early with respect to other cortical voxels (*Figure 5E*). An analogous, but opposite feature also appears in a subset of the cross-RSN sensorimotor (SMN) block, which assumes an 'all late' (red) configuration. In other words, this set of voxels becomes late with respect to cortical BOLD signal activity in nearly all other cortical voxels during SWS. The voxels in question correspond to the paracentral lobule (*Figure 2C*). That the paracentral lobule becomes late in SWS is also evident in *Figure 4C*. This effect is not significant in *Figure 5E* because the paracentral lobule is only part of the *a priori* defined SMN.

## Discussion

We investigated apparent propagation of spontaneous infra-slow activity, as measured by rs-fMRI, in wake and slow wave sleep. Our whole-brain comparison identified several cortical and subcortical regions exhibiting significantly different state-dependent lags (*Figure 2*). These regions were used for seed-based analyses. Lag maps seeded in the thalamus, brainstem, and putamen demonstrated that subcortical structures are early in relation to cortex during wake, but late during SWS (*Figure 3*). In contrast, propagation within the brainstem-thalamic axis was relatively preserved across states (*Figure 3*). Cortical seed-based lag maps, computed using visual cortex, medial prefrontal cortex, and paracentral lobule, exhibited complex, regionally specific effects (*Figure 4*). Finally, comparison of TD matrices in wake vs. SWS revealed that inter-RSN propagation was significantly altered in SWS whereas intra-RSN propagation was relatively preserved (*Figure 5C*).

These findings demonstrate dramatic rearrangements in propagated infra-slow intrinsic activity during SWS as compared to wake. By comparison, the effect of SWS on zero-lag correlation structure (functional connectivity) is much more modest (*Figure 6*). State differences in functional connectivity manifest as quantitative changes in the magnitude of correlations, for example, reduced correlations between anterior and posterior brain regions during SWS (*Picchioni et al., 2013*). However, the overall topography of resting state networks (RSNs) generally is preserved (*Horovitz et al., 2009*; *Larson-Prior et al., 2009*; *Picchioni et al., 2013*; *Sämann et al., 2011*; *Tagliazucchi et al., 2013*) (*Figure 6*). Interestingly, some of the present lag findings topographically correspond to effects observed with conventional functional connectivity. Specifically, SWS induces focal functional connectivity changes in visual cortex, paracentral lobule, and thalamus; this topography substantially overlaps some of the presently observed lag effects (compare sagittal views in *Figures 2C* and *6D, E*). In a similar vein, PET studies show that SWS, as well as pathological disorders of consciousness, both are associated with prominent reductions in thalamic metabolism (*Laureys, 2005*). This finding hypothetically corresponds, in the present results, to the prominent thalamic shift in lag status from early in wake to late in SWS. However, much more work is needed to determine whether these intriguing correspondences are in any way general.

### Relation of BOLD signal apparent propagation to electrophysiology

The implications of our findings critically depend on how spontaneous BOLD fMRI signal fluctuations relate to electrophysiology. This question currently remains a topic of active investigation (*Florin et al., 2015*). Several studies using invasive recordings have found that slow (0.5–4 Hz) and infra-slow (<0.1 Hz) local field potentials correspond most closely to BOLD signal fluctuations and correlation structure. This correspondence has been demonstrated in wake (*He et al., 2008*; *Hiltunen et al., 2014*; *Nir et al., 2008*), sleep (*He et al., 2008*), and anesthesia (*Pan et al., 2013*). Thus, a natural hypothesis is that temporal lags in the BOLD signal reflect propagation of slow and infra-slow electrophysiological activity. However, as far as we are aware, temporal lags in infra-slow electrophysiology have not been investigated at the systems level. On the other hand, propagation of slow activity has been widely reported. The spectral content of the slow oscillation at the low end, approximately 0.5 Hz, approaches the infra-slow range. Hence, in the following, we consider correspondence between reports of propagated slow electrophysiology and the present results, with the understanding that future investigations specifically of infra-slow propagation are needed.

Slow electrophysiological activity has been most studied in the context of SWS and anesthesia, during which the slow oscillation, or UP/DOWN states (UDSs), (*Steriade et al., 1993*) is a characteristic feature. UDSs classically are described as periodic at frequencies in the 0.5–1.5 Hz range, and are known to propagate on a time-scale of 100's of milliseconds (*Hahn et al., 2006*; *McCormick et al., 2015*; *Petersen et al., 2003*). More recently, UDSs have also been described in awake, resting rodents (*Ferezou et al., 2007*). Although UDS periodicity is a characteristic of single neuron membrane potential recordings (*Hahn et al., 2006*; *Petersen et al., 2003*), macro-electrode recordings, e.g., electroencephalography (EEG) and electrocorticography (ECoG), show aperiodic, 1/f-like spectral content during SWS (*He et al., 2010*; *Nir et al., 2011*; *Sirota et al., 2003*). More generally, the spectral content of intrinsic electrophysiologic activity and rs-fMRI is 1/f-like in wake as well as SWS (*He et al., 2008*; *Hiltunen et al., 2014*). Moreover, macro-electrode recordings of slow activity also show apparent propagation on time-scales as long as 1–2 s during sleep and anesthesia (*Nir et al., 2011*; *Sirota et al., 2003*), as in the present rs-fMRI data.

Propagated slow potentials during SWS share additional points of concordance with the present BOLD fMRI results. Electrophysiological studies in both humans and rodents suggest that slow activity tends to originate in medial prefrontal cortex and propagate to more posterior regions (*Massimini et al., 2004*; *Nir et al., 2011*; *Sheroziya and Timofeev, 2014*). We also observe a 'front-to-back' propagation pattern (see *Figures 4B and Figure 7—figure supplement 2*). However, this pattern should be understood as only one of many. In that regard, a noteworthy principle that has emerged on the basis of human high-density EEG and ECoG recordings is that slow waves originate in multiple locations and propagate through multiple routes (*Massimini et al., 2004*; *Murphy et al., 2009*; *Nir et al., 2011*). A necessary algebraic consequence of this principle is that the lag structure of propagated slow activity exists in a multi-dimensional space (*Mitra et al., 2015*). We have recently demonstrated precisely this point by analysis of a very large resting state fMRI dataset acquired in quietly resting awake adults (*Mitra et al., 2015*). We confirm that the same principle applies in the current fMRI data, both during wake and SWS (*Figure 7*). Indeed, the spatial principal components ('lag threads') derived by analysis of the fMRI time-delay matrix provides a compact means of visualizing how the propagation of intrinsic activity becomes rearranged in wake vs. SWS (*Figure 7—figure supplement 1* and *2*).

To the best of our knowledge, apparent propagation of either slow or infra-slow electrophysiological activity in the awake state has not been reported in either humans or animals. However, voltage sensitive dye (VSD) studies in quietly resting rodents have demonstrated multiple patterns of propagation of slow intrinsic activity over 100's of milliseconds (*Mohajerani et al., 2013*; *Mohajerani et al., 2010*). Moreover, propagation patterns assessed using VSD are bilaterally symmetric (*Mohajerani et al., 2010*), as are BOLD fMRI patterns of propagation (*Mitra et al., 2015*; *Mitra et al., 2014*).

In summary, the available evidence suggests that the BOLD signal and slow/infra-slow electrophysiological activity exhibit similarities in spectral content, correlation structure, and apparent propagation. The present results motivate future investigations to clarify the physiological links between propagated UDSs, propagated VSD patterns in rodents, and propagated BOLD fMRI signal fluctuations in humans, both during wakefulness and SWS. Direct electrophysiological studies of infra-slow propagation are especially needed. In the following, we discuss hypotheses regarding the possible functions of propagated infra-slow activity, observed using rs-fMRI, in relation to theories originally derived by observations of propagated slow potentials.

## Relation of present results to the physiology of slow wave sleep

Cortical re-arrangements in lag structure are complex and regionally dependent, as demonstrated by the seed-based analyses shown in *Figure 4*. In the following, we present three hypotheses, informed by prior studies of SWS, suggesting how infra-slow propagation may relate to the physiological functions of sleep. Future work combining rs-fMRI sleep recordings with behavioral data will be required to investigate these hypotheses.

First, slow wave sleep is believed to represent an off-line state during which slow activity enables consolidation of newly acquired memories (*Born et al., 2006*; *Maquet, 2001*; *Marshall et al., 2006*; *Stickgold, 2005*). Human imaging studies of word-pair association tasks have shown that the medial prefrontal cortex (mPFC) is implicated in consolidation of declarative memory (*Euston et al., 2012*; *Gais et al., 2007*; *Takashima et al., 2006*). Consolidation is accompanied by increases in mPFC activity during successful later recall (*Gais et al., 2007*; *Takashima et al., 2006*). The locus of these previously reported effects is notably close to the mPFC region illustrated in *Figure 2C* (cf. Figure 3 in *Takashima et al., 2006* and Figure 4 in *Gais et al., 2007*). Using the same task, Marshall and colleagues found that artificially initiating slow wave activity by applying current via frontal electrodes during SWS boosts subsequent recall (*Marshall et al., 2006*). Thus, the available data suggest that the mPFC shift towards earliness during SWS may facilitate consolidation of declarative memory. This hypothesis could be tested by relating infra-slow, front-to-back propagation to consolidation of declarative memory.

Second, SWS is also thought to improve procedural memory, e.g., motor sequence learning and motor adaptation (*Stickgold, 2005*). We observed significant wake vs. SWS latency shifts in putamen and paracentral lobule (PCL) (*Figures 2–4*), both of which structures are strongly associated with procedural learning (*Graybiel, 2005*; *Halsband and Lange, 2006*; *Knowlton et al., 1996*). The paracentral lobule is a functional component of the supplementary motor area (SMA) (*Lim et al., 1994*).

Sleep-dependent improvements in procedural tasks have been postulated to involve plasticity in cortico-striatal circuits (*Doyon and Benali, 2005*). Thus, the reversal in the temporal lag between putamen and cortex (*Figure 3B*) during SWS could be a correlate of off-line consolidation of procedural memory. Similarly, altered apparent propagation in PCL may relate to offline adjustment of motor programs. Alternatively, PCL lateness during SWS could represent a correlate of the abolition of motor behavior. These possibilities could be investigated by combining rs-fMRI studies of SWS with procedural learning paradigms.

Third, we show that visual cortex is neither wholly late nor early during wake, but that nearly the entire cerebral cortex becomes late with respect to visual cortex during SWS (*Figure 4A*, *Figure 5*). This visual cortex finding may relate to the now widely recognized fact that dreaming also occurs in NREM sleep (*Hobson et al., 2000*). Recent work has shown that BOLD fMRI activity in visual cortex predicts NREM dreaming (*Horikawa et al., 2013*). Hence, there may exist a link between initiation of infra-slow activity in visual cortex and dreaming. The question of whether initiation of infra-slow activity in visual cortices relates to dreaming has not been studied thus far, but could be explicitly examined using the methodology of Horikawa and colleagues (*Horikawa et al., 2013*).

## Relation of present results to thalamic gating of sensory input

Wakefulness is a state during which environmental awareness is possible (*Seth et al., 2005*). The thalamus plays a critical role in this state by relaying sensory signals to the cerebral cortex (*Alkire et al., 2008*). Electrophysiological recordings have shown that the awake state is associated with depolarization of thalamocortical cells, which facilitates transmission of ascending input to the cortex (*Brown et al., 2012*; *Franks, 2008*; *McCormick and Bal, 1997*). Conversely, during SWS, signals from the environment do not reliably reach the cortex owing to hyperpolarization of thalamocortical neurons (*Franks, 2008*; *Franks and Lieb, 1994*; *Hirsch et al., 1983*; *Steriade and Timofeev, 2003*). Thus, the thalamus 'gates' transmission of environmental stimuli to the cortex. Moreover, in vitro work suggests that thalamic hyperpolarization in SWS is modulated by propagation of slow (<1 Hz) activity from cortex to thalamus (*Blethyn et al., 2006*; *Hughes et al., 2002*). Our analysis of BOLD signal fluctuations is consistent with this account: Cerebral cortex leads thalamus during SWS, but thalamus leads cortex during wake (*Figure 3A*). Importantly, BOLD signal propagation from brainstem to thalamus is similar in wake and SWS. This result suggests that the thalamus is the site at which ascending signals are impeded during SWS, in accordance with the hypothesized role of the thalamus as a sensory gate.

## Relation of present findings to theories of consciousness

*Figure 5* shows that cross-network lag structure is extensively altered in SWS as compared to wake. During wake, both diagonal- and off-diagonal blocks of the TD matrix exhibit well-ordered early, middle, and late components. Thus, cross-network and within-network propagation are comparable during wake. During SWS, apparent propagation within and across networks is no longer comparable (*Figure 5B,E*). Cross-network (off-diagonal) blocks lose their 'early-to-late' structure, whereas within-network (diagonal blocks) lag structure is preserved (*Figure 5A-C*). Moreover, most cross-network blocks (excluding visual network pairs) appear disorganized during SWS. Speculatively, the dissociation between preserved within-network propagation and disorganized cross-network propagation suggests that neural communication during SWS, as measured by propagation of BOLD signals, is largely intact within functional networks (RSNs), but is disrupted across functional networks. These observations support theories postulating that reduced subjective awareness ('unconsciousness') during SWS results from loss of integration across networks (*Mashour, 2005*; *Tononi, 2004*). 'Conscious' here refers to the awake, or on-line, state. Accordingly, decreased subjective awareness during SWS theoretically results from loss of integration across networks, while within-network activity is generally preserved (*Boly et al., 2012*; *Mashour, 2005*; *Tagliazucchi et al., 2013a*). A corollary is that, during SWS, neural communication is maintained within functional modules, but altered across functional modules, resulting in 'network segregation' (*Mashour, 2013*; *Tagliazucchi et al., 2013a*). Our TD matrix results are consistent with this principle, provided the assumption that BOLD signal propagation reflects neural communication at a broad spatio-temporal scale,

We interpret our findings as follows. During wake, the brain is capable of responding to the environment and functional systems reciprocally communicate to sustain conscious content (*Alkire et al., 2008*; *McCormick et al., 2015*). Accordingly, we find ascending BOLD signal propagation from thalamus to cortex (*Figure 3A*), as well as well-ordered propagation of activity between resting state networks (*Figure 5A*). In SWS, off-line brain activity theoretically serves mechanisms concerned with synaptic homeostasis and memory consolidation (*McClelland et al., 1995*; *McCormick et al., 2015*). Features of the off-line state (SWS) include reduced responses to environmental stimuli, altered inter-network communication, but relatively intact intra-network communication (*McClelland et al., 1995*; McNaughton et al. 2003; *Tagliazucchi et al., 2013a*). Accordingly, we observe reversal of thalamo-cortical propagation (*Figure 3A*), altered cross-RSN propagation (*Figure 5B-C*), and intact within-RSN propagation. Future study of BOLD signal propagation and infra-slow activity is necessary to obtain a better understanding of the physiology of spontaneous activity in wake and sleep.

## Materials and methods

### EEG–fMRI acquisition and artifact correction

Acquisition parameters and details for these data have been previously published (*Tagliazucchi et al., 2013*). fMRI was acquired using a 3 T scanner (Siemens Trio) with optimized polysomnographic settings (1,505 volumes of T2*-weighted echo planar images, repetition time/echo time = 2,080 ms/30 ms, matrix = 64 × 64, voxel size = 3 × 3 × 2 mm$^3$, distance factor = 50%; field of view = 192 mm$^2$). 30 EEG channels were simultaneously recorded using a modified cap (EASY-CAP) with FCz as reference (sampling rate = 5 kHz, low pass filter = 250 Hz, high pass filter = 0.016 Hz). MRI and pulse artifact correction were performed based on the average artifact subtraction method (*Allen et al., 1998*) as implemented in Vision Analyzer2 (Brain Products) followed by ICA-based rejection of residual artifact components (CBC parameters; Vision Analyzer). EEG sleep staging was done by an expert according to the American Academy of Sleep Medicine (AASM) criteria (*Soeffing et al., 2008*).

### Subjects

63 non-sleep-deprived subjects were scanned in the evening (starting at ~8:00 PM). Subjects were instructed to keep eyes closed during wakefulness. Hypnograms were inspected to identify epochs of contiguous sleep stages lasting at least 5 min (150 volumes). These criteria yielded 39 subjects contributing to the present analyses. Included are 70 epochs of wakefulness, 47 epochs of N2 sleep, and 38 epochs of N3 sleep (SWS). Detailed sleep architectures of each participant have been previously published (*Tagliazucchi et al., 2013*).

### fMRI preprocessing

fMRI preprocessing was as described in (*Mitra et al., 2014*). Briefly, this included compensation for slice-dependent time shifts, elimination of systematic odd-even slice intensity differences due to interleaved acquisition, and rigid body correction of head movement within and across runs. Atlas transformation was achieved by composition of affine transforms connecting the fMRI volumes with the T2-weighted and T1-weighted structural images. Additional preprocessing in preparation for lags analysis included spatial smoothing (6 mm full width at half maximum (FWHM) Gaussian blur in each direction), voxel-wise removal of linear trends over each fMRI run, and temporal low-pass filtering retaining frequencies below 0.1 Hz. Spurious variance was reduced by regression of nuisance waveforms derived from head motion correction and timeseries extracted from regions (of 'non-interest') in white matter and CSF as well the BOLD timeseries averaged over the brain (*Fox et al., 2009*). Frame censoring was computed at a threshold of 0.5% root mean square frame-to-frame intensity change (*Power et al., 2012*). Epochs containing fewer than 10 contiguous frames were excluded. These criteria removed 5.3 ± 0.9% of frames per individual during wake, 6.1 ± 0.7% during N2 sleep, and 5.8 ± 0.7% during N3 sleep. There were no statistically significant differences in the amount of frame censoring by state.

## Computation of lag between BOLD time series

Our method for computing lags between time series has been previously published (*Mitra et al., 2014*). Conventional seed-based correlation analysis involves computation of the Pearson correlation, $r$, between the time series, $x_1(t)$, extracted from a seed region, and a second time series, $x_2(t)$, extracted from some other locus (single voxel or region of interest). Thus,

$$r_{x_1 x_2} = \frac{1}{\sigma_{x_1} \sigma_{x_2}} \frac{1}{T} \int x_1(t) \cdot x_2(t) dt, \tag{1}$$

where $\sigma_{x_1}$ and $\sigma_{x_2}$ are the temporal standard deviations of signals $x_1$ and $x_2$, and $T$ is the interval of integration. Here, we generalize the assumption of exact temporal synchrony and compute lagged cross-covariance functions. Thus,

$$C_{x_1 x_2}(\tau) = \frac{1}{T} \int x_1(t + \tau) \cdot x_2(t) dt, \tag{2}$$

where τ is the lag (in units of time). The value of τ at which $C_{x_1 x_2}(\tau)$ exhibits an extremum defines the temporal lag (equivalently, delay) between signals $x_1$ and $x_2$ (*König, 1994*). Although cross-covariance functions can exhibit multiple extrema in the analysis of periodic signals, BOLD time series are aperiodic (*He et al., 2010*; *Maxim et al., 2005*), and almost always give rise to lagged cross-covariance functions with a single, well defined extremum, typically in the range ±1 s. We determine the extremum abscissa and ordinate using parabolic interpolation (*Mitra et al., 2014*).

Given a set of $n$ time series, $\{x_1(t), x_2(t), \cdots, x_n(t)\}$, finding all $\tau_{i,j}$ corresponding to the extrema, $a_{i,j}$, of $C_{x_i x_j}(\tau)$ yields the anti-symmetric, time delay matrix:

$$TD = \begin{bmatrix} \tau_{1,1} & \cdots & \tau_{1,n} \\ \vdots & \ddots & \vdots \\ -\tau_{1,n} & \cdots & \tau_{n,n} \end{bmatrix} \tag{3}$$

The diagonal entries of $TD$ are necessarily zero, as any time series has zero lag with itself. Moreover, $\tau_{i,j} = -\tau_{j,i}$, since time series $x_i(t)$ preceding $x_j(t)$ implies that $x_j(t)$ follows $x_i(t)$ by the same interval. Here, the timeseries were extracted from $(6mm)^3$ cubic voxels evenly distributed throughout gray matter in the whole brain (*Mitra et al., 2015*).

We projected the multivariate data represented in the $TD$ matrix onto one-dimensional maps using the technique described by Nikolic and colleagues (*Nikolić, 2007*; *Schneider et al., 2006*). We refer to these one-dimensional maps as lag projections. Operationally, the projection is done by taking the mean across the columns of $TD$ (*Equation 3*), that is,

$$T_p = \left[ \sum_{j=1}^{n} \tau_{1,j} \quad \cdots \quad \sum_{j=1}^{n} \tau_{n,j} \right] \tag{4}$$

Seed-based lag maps were computed according to *Equation 2*, except instead of computing lags between all voxels, we computed the lag between a reference timeseries extracted from the seed and timeseries extracted from all $(6mm)^3$ voxels. This procedure produces a one-dimensional seed-based lag map.

Group level $TD$ matrices, lag projections, and seed-based lag maps were obtained in each state (W, N2 sleep, and N3 sleep) by computing each quantity at the individual subject level (averaging across temporally contiguous epochs) and then averaging.

## Statistical analysis

Statistical significance of wake versus SWS differences in lag projection maps was assessed on a cluster-wise basis using threshold-extent criteria computed by extensive permutation resampling (*Hacker et al., 2012*; *Hayasaka and Nichols, 2003*). *Figure 5C* reports Spearman $\rho$ correlations over entries in $TD$ matrix blocks in wake and SWS. Statistical significance was assessed by permutation resampling of the matrix entries within blocks, with correction for multiple comparisons. *Figure 5D and E* report mean lag values averaged over entries in $TD$ matrix blocks. Statistical significance of the difference in mean lag value across TD matrix blocks was also assessed through permutation resampling, with correction for multiple comparisons.

## Acknowledgements

This work was supported by the National Institute of Health (NS080675 to MER and AZS; P30NS048056 to AZS; F30MH106253 to AM), the Bundesministerium für Bildung und Forschung (grant 01EV0703) and the LOEWE Neuronale Koordination Forschungsschwerpunkt Frankfurt (NeFF).

## Additional information

### Funding

| Funder | Grant reference number | Author |
| --- | --- | --- |
| National Institutes of Health | NS080675, P30NS048056, F30MH106253 | Anish Mitra<br>Abraham Z Snyder<br>Marcus E Raichle |
| Bundesministerium für Bildung und Forschung | 01EV0703 | Enzo Tagliazucchi<br>Helmut Laufs |
| LOEWE Zentrum AdRIA | | Enzo Tagliazucchi<br>Helmut Laufs |

The funders had no role in study design, data collection and interpretation, or the decision to submit the work for publication.

### Author contributions

AM, Conception and design, Analysis and interpretation of data, Drafting or revising the article; AZS, MER, Analysis and interpretation of data, Drafting or revising the article; ET, HL, Conception and design, Acquisition of data, Drafting or revising the article

### Ethics

Human subjects: Written informed consent was obtained from all subjects whose data was analyzed in this study, and data collection for this study was approved by the Goethe University ethics committee.

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
