## [Decision Letter]

Thank you for submitting your work entitled "Propagated slow intrinsic brain activity reorganizes across wake and slow wave sleep" for consideration by *eLife*. Your article has been reviewed by three peer reviewers, one of whom, Heidi Johansen-Berg, is a member of our Board of Reviewing Editors, and the evaluation has been overseen by the Reviewing Editor and the Senior Editor (Eve Marder). One of the reviewers, Marcello Massimini, has agreed to reveal his identity.

The reviewers have discussed the reviews with one another and the Reviewing editor has drafted this decision to help you prepare a revised submission.

Summary:

The manuscript by Mitra and colleagues thoroughly addresses a relevant issue regarding intrinsic brain oscillations. By means of a recently developed methodology (Mitra et al., 2014), the authors show that the temporal lag structure (apparent propagation) of rs-fMRI derived BOLD signal markedly changes from wakefulness to slow wave sleep (SWS). Overall, during SWS, ordered propagation (typical of wakefulness) is preserved within traditionally defined resting state networks (RSNs) but lost between RSNs. Additionally, propagation between cerebral cortex and subcortical structures reverses directions, and intra-cortical propagation becomes reorganized, especially in visual and sensorimotor cortices. The results are discussed in light of hypotheses regarding sleep function. The reviewers found this paper interesting, important and well conducted.

Essential revisions:

1) Did subjects have their eyes open or closed during wakefulness? This is a key point for interpreting any differences between sleep and wake, particularly given the dramatic changes found in visual cortex. This should be specified in the manuscript. If eyes were open during wake this may raise serious questions over the extent to which sleep and wake can be compared.

2) We recognise the challenges of recording REM sleep in the scanner. Did REM episodes occur during any of the sessions? The authors may want to comment on the expected changes in the temporal lag structure during REM sleep, in particular regarding the visual and thalamic regions. It may be relevant in the context of the proposed link of the observed findings with both sleep function and consciousness state. Specifically, would the cortico-subcortical lag be more similar to wake or slow wave sleep?

3) Are the RSNs chosen similarly in the wake and sleep conditions? Otherwise the differences between the two states may result in part from the RSN distribution rather than from the speed of progression within the RSNs. This is relevant, because while the RSN's may maintain general appearance in sleep as compared to wake, the size/distribution of the network does change as shown by earlier studies and also here in Figure 6. Of course, this is a bit of a chicken-and-egg problem, as the networks may appear different in conventional RSN analyses due to changes in timing of the constituent voxels, or the timing between these voxels may change because the networks change in appearance.

4) The section on memory consolidation is too speculative given that no behavioural data were acquired. This section should be removed. Potential relevance to consolidation could be briefly mentioned but the lack of behavioural data in the current study should also be acknowledged as a limitation.

5) Given the absence of a common behavioural experience for the participants, the results need to be understood separately from such consolidation; for example, the conspicuous change in the timing of the visual cortex is striking and not explained in terms of its possible function apart from possible processing of visual discrimination tasks (which we do not find a convincing explanation given that no such tasks were employed here).

---

## [Author Response]

Essential revisions:

*1) Did subjects have their eyes open or closed during wakefulness? This is a key point for interpreting any differences between sleep and wake, particularly given the dramatic changes found in visual cortex. This should be specified in the manuscript. If eyes were open during wake this may raise serious questions over the extent to which sleep and wake can be compared.*

Subjects were instructed to have their eyes closed at all times, including during wakefulness; thus, the visual cortex changes are not attributable to an eyes open vs. eyes closed effect. This important methodological point is now made explicit both in the Methods and Introduction:

“Subjects were instructed to keep eyes closed during wakefulness.”

“Here, we report whole-brain patterns of propagated rs-fMRI activity in humans, contrasting eyes-closed wake vs. SWS. Our results lend support to each of the aforementioned perspectives.”

*2) We recognise the challenges of recording REM sleep in the scanner. Did REM episodes occur during any of the sessions? The authors may want to comment on the expected changes in the temporal lag structure during REM sleep, in particular regarding the visual and thalamic regions. It may be relevant in the context of the proposed link of the observed findings with both sleep function and consciousness state. Specifically, would the cortico-subcortical lag be more similar to wake or slow wave sleep?*

As noted, achieving REM in a MRI scanner is very difficult and usually requires sleep deprivation. In healthy subjects, REM does not occur within the first hour of sleep. Scanning for longer than one hour is challenging both for the subjects and in terms of scanner stability and eventually data quality. Our objective was to collect high quality data during “natural sleep” (as much as it is possible given the experimental condition). Hence, we did not sleep-deprive our subjects and, in fact, there were no REM sleep epochs in any session. Two of the authors (ET and HL) are presently conducting a study on sleep-deprived subjects in an attempt to collect REM data. We would like to study lag structure during REM sleep and may be able to do so on the basis of this ongoing study. The electrophysiology and neuromodulator tone of REM sleep resembles wakefulness more than SWS (Brown et al., 2012). Accordingly, we might predict that the cortico-subcortical lag during REM sleep will resemble wake more than SWS. However, at present, this remains mere speculation. The obvious behavioral differences between REM sleep and wake could support an opposite prediction. In conclusion, we cannot speculate on this topic in the present manuscript.

*3) Are the RSNs chosen similarly in the wake and sleep conditions? Otherwise the differences between the two states may result in part from the RSN distribution rather than from the speed of progression within the RSNs. This is relevant, because while the RSN's may maintain general appearance in sleep as compared to wake, the size/distribution of the network does change as shown by earlier studies and also here in Figure 6. Of course, this is a bit of a chicken-and-egg problem, as the networks may appear different in conventional RSN analyses due to changes in timing of the constituent voxels, or the timing between these voxels may change because the networks change in appearance.*

The same RSN definitions were applied during wake and sleep; RSN topographies are shown in Figure 5—figure supplement 2. This important point has been clarified in the revision: “The same RSN definitions are applied in wake and SWS.”

The Figure 6 caption states that the voxels in the correlation matrices are sorted into RSNs as in Figure 5. Thus, this revision now also makes it clear that the correlation matrices in Figure 6 are arranged identically in wake and SWS.

With respect to the chicken-and-egg problem, we reason as follows. We have previously shown that RSNs (zero-lag correlations) can arise on the basis of lag structure but not vice versa (Mitra et al., 2015). Here, we show major changes in lag structure across wake and SWS (Figure 2, Figure 3 and Figure 4) with only minor changes in correlation structure (Figure 6). Indeed, in Figure 6, diagonal blocks of high correlation, representing RSNs, are nearly unchanged. Even off-diagonal blocks, for instance, the anti-correlation between the DMN and dorsal attention network (DAN), are generally preserved. Quantitatively, if we compare all unique values in the wake and SWS correlation matrices in Figure 6, we find that the Pearson correlation across states is r = 0.89. This finding is in line with prior work demonstrating that the spatial topography of RSNs is only minimally altered across wake vs. SWS (Picchioni et al., 2013).

We still have to account for the observation of major changes in lag structure but only minor changes is correlation structure. Whereas Figure 2, Figure 3, Figure 4 and Figure 6 do not rely on RSN definitions, Figure 5 and Figure 6AB do. Now, taking the DMN:DAN block as an example, the extent of alteration in lag structure (Figure 5, yellow squares) in this block is not explainable by the minimal change in correlation structure in this same block (Figure 6AB). According to our prior results (Mitra et al., 2015), it is likely that the changes in lag structure (shifts on the order of a second) underlie the observation of subtle functional connectivity changes. As the spectral content of BOLD fluctuations lies in the infra-slow range (<0.1 Hz, i.e., periods on the order of 10’s-100’s of seconds), it is entirely consistent to observe only small changes in zero-lag correlations in response to temporal shifts on the order of one second. Moreover, this perspective explains the general preservation of RSN structure (Figure 6) on the basis of the preservation of *intra-RSN* lag structure in wake vs. SWS (Figure 5). However, as the present study lacks an experimental or computational basis for relating functional connectivity to temporal structure, we have elected to omit these points from the main text.

*4) The section on memory consolidation is too speculative given that no behavioural data were acquired. This section should be removed. Potential relevance to consolidation could be briefly mentioned but the lack of behavioural data in the current study should also be acknowledged as a limitation.*

We agree that the section relating the present results to memory consolidation was overly speculative. That text has been removed. However, we do believe that it is important to discuss the present results in the context of prior work on the physiological functions of spontaneous activity during SWS. Even though we cannot directly link the current findings to behavioral measures, we believe that the significance of our findings is informed by prior electrophysiological studies of SWS. Accordingly, the section on memory consolidation has been re-cast as a discussion of hypotheses regarding how our findings may relate to the physiology of slow wave sleep. The revised Discussion touches on memory consolidation, immobility during sleep, and dreaming. We explicitly note the absence of behavioral data, and state that future studies combining sleep rs-fMRI recordings with behavioral tasks will be required to test stated hypotheses. We hope that the revised text (subheading “Relation of present results to the physiology of slow wave sleep”) appropriately relates our findings to prior work without veering into unwarranted speculation.

*5) Given the absence of a common behavioural experience for the participants, the results need to be understood separately from such consolidation; for example, the conspicuous change in the timing of the visual cortex is striking and not explained in terms of its possible function apart from possible processing of visual discrimination tasks (which we do not find a convincing explanation given that no such tasks were employed here).*

As recommended, the section relating visual cortex earliness to perceptual memory consolidation mechanisms has been removed. Instead, we now discuss the visual cortex finding in terms of a possible relation to NREM dreaming, as quoted above. Importantly, this discussion text is explicitly framed as hypothetical (paragraph four, subheading “Relation of present results to the physiology of slow wave sleep”).

Although studies of brain function often employ behavioral paradigms to isolate specific patterns of activity, we suggest that the conspicuous change in the timing of visual cortex activity in the absence of a common manipulation is consistent with a consolidation mechanism. For example, anterior-posterior propagation of slow activity is a feature of SWS, even in the absence of a common behavioral manipulation (Massimini et al.. 2004). This prominent pattern of propagated slow activity has been previously related to the consolidation of declarative memory (Marshall et al.. 2006). A mechanism supporting consolidation generally (e.g., anterior-posterior propagation of slow activity) may not depend on the specific content of the material being consolidated. The same could be said about visual cortex earliness as a correlate of perceptual memory consolidation.

Consolidation is a well-recognized function of SWS. Accordingly, some discussion of this topic, as detailed above, seems appropriate. However, as the presently analyzed data were acquired to investigate sleep, but not specifically consolidation, this discussion should be limited. We hope that the revised text outlining testable hypotheses is appropriately scaled.